# The Effect of Clay on the Shear Strength of Microbially Cured Sand Particles

**DOI:** 10.3390/ma15103414

**Published:** 2022-05-10

**Authors:** Deluan Feng, Haiqin Gao, Zhanlin Li, Shihua Liang

**Affiliations:** School of Civil and Transportation Engineering, Guangdong University of Technology, Guangzhou 510006, China; wolfluan@126.com (D.F.); a13129386086@163.com (H.G.); 2112109084@mail2.gdut.edu.cn (Z.L.)

**Keywords:** viscous particle content, microbial curing, strength characteristics, triaxial consolidation without drainage test

## Abstract

Microbial solidification of sand has obvious effects: energy-saving and environmental protection. It is a green and sustainable soil consolidation technology with low energy consumption, which meets the needs of high-quality development of modern economy and society. However, when clay is doped in sand, clay has an uncertain influence on the effectiveness of the microbial solidification of sand. Therefore, triaxial consolidation undrained tests before and after microbial solidification of sands with different clay content are carried out in this paper. The effects of clay content on the solidification effect of sands are compared and analyzed. The variation laws of shear strength, unconfined compressive strength, internal friction angle and the cohesion of sands with different clay content before and after microbial solidification are discussed. The failure modes of sand samples were studied and the influence mechanism of clay on the microbial solidification of sand was revealed from a micro perspective. The test results show that the failure strain and unconfined compressive strength of microbial-induced calcium carbonate precipitation (MICP) treated samples increase first and then decrease with the increase in the clay content. The unconfined compressive strength is the highest when the clay content is 9%, and the samples with low clay content (3~9%) can still retain good integrity after being destroyed. As the content of clay in the sand–clay mixture increases, the internal friction angle of the sample decreases and the cohesion increases. After MICP treatment, the internal friction angle and cohesion of the sand increase first and then decrease with the increase in clay content. There are three main contact modes between sand-clay-CaCO_3_. When clay content is low, clay plays a filling role. The contact mode between sand-clay and CaCO_3_ is mainly between sand particles and calcium carbonate and between clay particles and calcium carbonate. When clay content is high, the contact mode between particles is mainly between clay particles and calcium carbonate. Higher clay content wraps sand particles, prevents contact between calcium carbonate and sand particles and reduces the strength of sand.

## 1. Introduction

Microbial-induced calcium carbonate precipitation technology has the advantages of green environmental protection, low cost, an adjustable reaction rate, and simple construction. It is a green and sustainable technology with low energy consumption and low emission [1,2]. The microbial bacterial solution in this technology has good fluidity during the construction process, and its own metabolites can induce the formation of calcium carbonate precipitation with a strong cementation ability. Therefore, it has been widely used in major geotechnical engineering fields, such as solidification of loose soil, foundation improvement, repair of concrete cracks, anti-seepage plugging, and anti-corrosion [3,4].

Researchers in the world have carried out a lot of research around this energy-saving and environment-friendly green geotechnical reinforcement technology, especially in the reinforcement of sand, and achieved fruitful research results. Given the good improvement effect of MICP technology in civil engineering, experts and researchers have begun to introduce this technology into the research on sand and achieved fruitful results. The unconfined compressive strength of Ottawa silica sand treated by MICP is similar to that of the sample treated with 10% cement and much better than that of lime treatment. The sand particle morphology and bonding mechanism play a key control in MICP grouting and significantly affect the cementation structure and overall strength after curing [5,6]. The effect of MICP curing sand particles with different particle sizes is obviously different. It has been found that the unconfined compressive strength of artificial silica sand and Ottawa sand is the largest when the particle size is 25–40 mesh [7]. Other researchers have studied the stress–strain behavior of MICP solidified sand. After MICP treatment, the shear strength of sand increases with the increase in MICP cementation, and the increase in cementation changes the strain hardening of sand into strain softening [8]. In addition, the permeability of sand after MICP treatment is significantly reduced, and the liquefaction resistance is significantly improved [9,10].

These research results show that MICP technology has achieved good results in solidifying sand. However, the sand layers in some areas of China are often rich in clay particles. Previous studies have found that the existence of clay particles has a serious impact on the physical and mechanical parameters of sand, especially shear strength, disintegration, etc., in which clay results in a significant reduction in shear strength of sand [11,12]. Kaolin can cause significant disintegration of granite residual soil [13,14]. Therefore, the presence of clay significantly weakens the physical and mechanical properties of sand such as shear strength and water stability, which may have an impact on MICP solidified sand.

Therefore, large number of researchers have carried out exploratory research on the effect of clay particles on microbial solidification of sand and obtained considerable achievements. Sun [15] performed an indoor physical model experimental study on MICP to improve solidified sand and compared and analyzed the influence of different clay contents on the solidification effect. The results showed that the existence of clay, such as kaolin, inhibits the urease activity of bacteria. As a result, the permeability of the sand column decreases rapidly with the increase in kaolin content. Ma et al. [16] mixed different concentrations of a bentonite solution into coarse sand and conducted an experimental study on indoor MICP solidification. The findings revealed that the addition of bentonite greatly reduces the permeability and strength of coarse sand mainly because the addition of bentonite blocks the pores of the sample, interrupts the flow path of the bacterial solution, and leads to the discontinuity of calcium carbonate cement. Yue et al. [17] studied the effect of mature rice pulp on the MICP and solidification of sediments in the Yellow River Basin and found that mature rice pulp can improve bacterial activity and promote the formation of calcium carbonate. Ma et al. [18] found that the main function of kaolin is to assist nucleation and increase the amount of effective calcium carbonate precipitation, so as to enhance the effect of curing coarse sand. Jiang et al. [19] studied the internal erosion control of MICP on seepage induced by sand-clay mixture. The main reason was that the carbonate precipitation generated reduced the permeability of sand mixture and thus improved the erosion resistance of sand. The abovementioned researchers have conducted extensive research on the changes in bacterial urease activity, calcium carbonate production, and sand column permeability in the process of microbial sand solidification. They found that the addition of clay greatly influences the solidification effect of MICP.

However, these studies rarely involved the investigation of the influence of clay content on the strength characteristics of microbial solidified sand. Mixtures of sand and clay exist widely in South China. Clay significantly reduces the shear strength of sand, thus affecting the pile quality of microbial sand piles and even leading to the damage of sand piles [20,21,22]. In this study, Fujian standard sand is used as the raw material. Clay is added to the sand to manually prepare a standard sample, and unconfined compressive strength and consolidated undrained triaxial tests are conducted before and after MICP curing treatment. The influence mechanism of different clay contents on the microbial curing effect is analyzed through scanning electron microscopy testing technology. The research results provide scientific support and an experimental basis for the application of microbial sand piles in South China.

## 2. Test Materials and Schemes

### 2.1. Test Materials

The clay used in this test is from Dayong Town, Nansha District, Guangzhou, and has a soil depth of about 5 m. The water content is 69.28%, and the liquid and plastic limits are 52.84% and 38.14%, respectively. The basic physical indexes of the clay particle are shown in Table 1. The particle size distribution of the clay measured by the laser particle size analyzer is shown in Figure 1. As shown in Figure 2, the mineral composition of clay is mainly montmorillonite, illite, kaolinite and quartz. The focus of this paper is on the effect of clay particles on the strength properties of microbially cured sandy soils. The initial grouting conditions keep unchanged for each MICP treatment sample. The test sand is Fujian Xiamen ISO standard sand, which is fine sand with poor particle grading. The physical characteristic parameters are shown in Table 2, and the particle grading is shown in Figure 3. According to USCS, the classification of clay and sand are CH and SP, respectively [23,24].

The strain used in the experiment is Pasteurella octopus, which is the most commonly used strain in the research field of MICP. It is purchased from DSM Company in the Helen, The Netherlands (number DSM33). The strain can rapidly decompose urea, produce a large number of carbonate ions, and attach to the surface of sand particles, thus providing an attachment point for the formation of calcium carbonate.

Different content of clay is added to the sand to form a sand-clay mixture which is then cured by MICP. The clay content is 0, 3%, 6%, 9%, 12%, 15%, 18% and 21%, respectively. It should be noted that the change of the clay content not only changes the water content but also the unsaturated behavior of soil samples [25,26]. However, the present work mainly focuses on the influence of clay content on the solidification effect of MICP treated sand, and as soil samples with different clay content undergo MICP grouting procedure, which inevitably causes changes to the water content and unsaturated behavior of soil samples, therefore, the unsaturated was not considered.

### 2.2. Test Scheme

The test scheme is shown in Table 3. In the table, NL0 to NL21 refer to three parallel samples in each group for the consolidated undrained (CU) test. NLC0 to NLC21 refer to six parallel samples in each group, namely, three for the unconfined compressive strength test and three for the CU test. The loading rate during the shear process is 0.9 mm/min, and the termination test strain is 20%. Each group of tests is conducted according to the relevant soil test specification [27].

## 3. Analysis of Test Results

### 3.1. The Influence of Clay Content on the Stress–Strain Curve

Figure 4 shows the stress–strain curves of sand–clay mixed soil with different clay contents after MICP treatment. When the sand column is subjected to axial pressure, its axial stress increases slowly. When the strain reaches the failure value, the stress decreases sharply then tends to stabilize. The stress–strain curves of the different clay contents differ but can be divided into four stages.

Stage 1: A small range of strain variation and sharp increase in stress. The interaction between calcium carbonate and sand particles connects the sand column to the whole through the “connecting bond” between calcium carbonate and sand particles to bear the pressure together. At this stage, the variation range of the strain is small, and the stress increases greatly;

Stage 2: A large strain range and slow increase in stress. At this time, the sample is subjected to a large bearing capacity, and some “connecting keys” between calcium carbonate and sand particles begin to break, but they can continue to bear the great pressure until the “connecting keys” inside the sand column break and reach the peak stress. At this stage, the main representations are as follows: the variation range of strain is large, and the increase range of stress is small;

Stage 3: A small strain range and sharp decrease in stress. At this time, the “connecting key” in the sand column is broken, so the column can no longer bear the peak stress and brittle failure. At this stage, the variation range of the strain is small, and the stress decreases greatly;

Stage 4: A large strain range and gentle reduction in stress. At this time, due to the bearing capacity of the sand column, the sand column is compacted, and the pores in it are filled tightly, resulting in the flattening of the stress curve. At this stage, the main representations are as follows: the variation range of strain is large, and the reduction range of stress is small.

Figure 4 shows that the strain of the sand columns with different clay contents differs obviously during failure. The failure strain increases initially then decreases with the increase in clay content. The failure strain of the sand column is the largest when the clay content is 9% and the smallest when the clay content is 21%. This result shows that the sample of the NLC9 group has the largest plasticity, better deformation ability, and better resistance to the deformation caused by external forces compared with the samples in the other groups. When the clay content increases from 12% to 21%, the brittle failure of the sample becomes increasingly obvious, and the ability to resist deformation is getting worse and worse. The stress decreases sharply after reaching the peak value, the sample begins to crack, and the crack develops rapidly. This condition is unfavorable to actual projects and should be avoided.

### 3.2. The Effect of Different Clay Contents on Unconfined Compressive Strength

An unconfined compressive strength test is conducted on the NCL0–NCL21 groups of samples (three samples in each group). The unconfined compressive strength and failure strain corresponding to different clay contents are obtained and shown in Figure 5.

With the increase in clay content, the unconfined compressive strength increases initially then decreases. The unconfined compressive strength of NLC3, NLC6, NLC9, and NLC12 groups was 1.28, 1.55, 1.90, and 1.46 times higher than that of the NLC0 group, respectively. The unconfined compressive strength of NLC15, NLC18, and NLC21 groups is lower than that of NLC0, indicating that when the clay content is greater than 15%, excessive clay exerts an adverse impact on the curing effect of microorganisms. The failure strains of NLC0, NLC3, NLC6, NLC9, NLC12, NLC15, NLC18, and NLC21 are 2.24%, 2.42%, 3.17%, 3.54%, 2.24%, 1.86%, 1.12%, and 0.99%, respectively, and present a trend of increasing initially then decreasing. When subjected to external load, the compressive strength of the dense sand sample is high because the movement and deformation between particles in the dense sand sample are limited, which improves the strength and failure strain of the sample. The results on unconfined compressive strength and failure strain show that in the experiment on the effect of clay content on the curing process of MICP, the unconfined compressive strength is the highest when the clay content is 9%. The curing effect and strength of the specimens can be improved by 3~12% viscous content compared with those without viscous particles. Higher clay content is detrimental to the curing process and reduces the strength of the sand.

### 3.3. The Effect of Different Clay Contents on the Failure Mode of the Sample

The sample after the unconfined compressive strength test is examined, and its damaged shape is shown in Figure 6.

Figure 6 indicates that when the clay content is 0%, the sand particles are peeled off after the sample is damaged and become loose after damage. When the clay content is 3%, 6%, and 9%, the sample can still maintain good integrity after damage, and when the clay content is 9%, its integrity is better than that when the clay content is 3% and 6%. When the clay content is 12%, 15%, and 18%, after the sample is damaged, the sample can be divided into several parts, and the integrity is poor. The cracks increase and widen with the increase in clay content. When the clay content is 21%, the integrity of the sample after damage is strong according to the apparent diagram of the sample. However, a fracture begins to appear in the middle of the sample because of the high content of clay particles. Excessive clay particles are deposited at the bottom of the sample with the flow of bacterial and nutrient solutions to inhibit the solidification effect of microorganisms. The content of effective calcium carbonate generated at the bottom of the sample is small. When the sample is loaded, a sliding failure occurs quickly between sand and clay particles at the bottom, resulting in a reduction in unconfined compressive strength.

The calcium carbonate induced by microorganisms plays a connecting role when the sample is subjected to external load. When abundant calcium carbonate is present, the sample can be connected as a whole and can still maintain good integrity when damaged.

### 3.4. The Effects of Different Clay Contents on Cohesion and Internal Friction Angle under the Consolidated Undrained Test

Figure 7 shows the principal stress difference of the different clay contents obtained from the consolidated undrained test on NL0–NL21 groups of samples (σ1–σ3 maximum failure point strength curve). When the clay content is the same, the greater the confining pressure is, the greater the peak failure point strength of the sample is. Under the same confining pressure, with the increase in clay content, the peak strength of sand–clay mixed soil gradually decreases, indicating that the increase in clay content reduces its shear strength.

Figure 8 shows the variation curve of the cohesion and internal friction angle of NL0–NL21 samples with the increase in clay content. The internal friction angle of sand–clay mixed soil (φ’) decreases with the increase in clay content, and the cohesion value (c’) increases with the increase in clay content. When the clay content increases from 0 to 21%, the cohesion increases from 2.5 kPa to 12.1 kPa, increasing by 384%, while the internal friction angle decreases from 36.5° to 24.18°. The higher the clay content is, the gentler the rising trend of cohesion is. Figure 7 and Figure 8 show that the change trend of the peak failure strength of the sample is similar to that of the internal friction angle, indicating that the shear strength of sand–clay mixed soil is mainly determined by the internal friction angle at this time.

The Cu test is also performed on the NLC0–NLC21 groups of samples (the remaining three samples in each group) to obtain the cohesion and internal friction angle under different clay contents and compare them with those of the sample without MICP treatment.

Analysis of Figure 9 reveals that internal friction angle φ’ and cohesion C’ increase initially then decrease with the increase in clay content. They reach the maximum when the clay content is 9%. Comparing with Figure 8 and Figure 9, when the clay content is the same, the internal friction angle after MICP treatment is not much higher, but the cohesion is significantly increased. When the clay content is less than 9%, the increase in internal friction angle after MICP treatment is less than 30%. When the clay content exceeds 9%, the maximum increase is 44.22%. When the clay content is 9%, the internal friction angle increases by more than 50–50.18%. Moreover, when the clay content is 21%, the increment in cohesion after MICP treatment is the smallest (from 12.1–129 kPa), and the maximum increment reaches 257.5 kPa at 9% clay content, indicating an increase of 34.33 times. A significant relationship is observed between the increase in internal friction angle and the contact form between particles. At the beginning, the contact between the particles of the specimens that are not treated and treated by MICP is mainly the contact between sand particles. With the increase in clay content, the contact form of the former becomes the contact between sand and clay particles, whereas that of the latter becomes the contact between sand particles and calcium carbonate. Among the three contact types, the friction resistance between sand particles and calcium carbonate is the highest. The friction resistance of sand particles in contact with sand particles ranks in the middle. Sand particles have the smallest contact with clay particles, but the difference between them is small. Therefore, the increase in the internal friction angle after MICP treatment is not obvious. However, after MICP treatment, the mechanical biting force between the particles of the sample is significantly increased, and the cohesion is greatly improved. The specimens can then withstand high external forces before shear failure occurs. Therefore, the shear strength of the specimens without MICP treatment is significantly improved.

## 4. Mechanism Analysis of the Influence of Clay Content on Microbial Curing

### 4.1. Mechanism Analysis of the Influence of Different Clay Contents on Unconfined Compressive Strength

Obvious differences in unconfined compressive strength and failure strain are observed at different clay contents. After MICP treatment, three main contact modes are observed among sand, clay, and calcium carbonate: clay and calcium carbonate (contact point 1 in Figure 10), sand and calcium carbonate (contact point 2 of Figure 10), and clay and sand (contact point 3 of Figure 10). When the clay content is low, the clay is mainly used for fill, and the contact mode among sand, clay, and calcium carbonate is mainly contact points 1 and 2. When the clay content is high, the contact mode among sand, clay, and calcium carbonate is mainly contact points 1 and 3.

A low clay content can improve the strength of sand for the following reasons:

(1)Clay particles are evenly distributed in the pores of the sand column, thereby increasing the specific surface area of the sand particles and providing good “nuclear sites” and good living space for bacteria to produce abundant effective calcium carbonate;(2)Bacteria are negatively charged, whereas clay particles are positively charged so that they can better adsorb bacteria. The injected bacteria are retained in the sand column to avoid bacterial loss and generate abundant effective calcium carbonate to improve the strength of the sand column;(3)The contact mode among sand, clay, and calcium carbonate is mainly contact points 1 and 2. When subjected to load, contact points 1 and 2 can provide good friction resistance to prevent sample failure and improve sample strength. Meanwhile, calcium carbonate can limit the movement and deformation between sand particles and improve the unconfined compressive strength and failure strain of the sample.

### 4.2. Analysis of the Influence Mechanism of Different Clay Contents on Cohesion and Internal Friction Angle

For the NL0–NL21 group samples (without MICP treatment), the influence of clay on mixed soil is mainly related to the formation of the soil matrix membrane. After clay and sand particles are mixed, three forms of contact exist between particles: contact between sand particles, contact between sand and clay particles, and contact between clay particles. The contact between sand and clay particles forms a matrix that wraps sand particles with a certain thickness, which then becomes a “soil matrix film”, as shown in Figure 11.

When the clay content in the mixed soil is low (3–9%), the clay fills the pores between sand particles. At this time, the contact form is mainly the contact between sand particles. The internal friction angle of the mixed soil is large, the cohesion is low, and the shear strength is high. With the increase in clay content, the pores between sand particles are filled, and a soil matrix film that wraps the sand particles is gradually formed. The contact form between particles changes from the contact between sand particles to the contact between sand and clay particles. The internal friction angle decreases, the cohesion increases, and the shear strength decreases gradually. At this time, the content of clay particles can continue to increase when these particles are in full contact with the matrix. The internal friction angle of the mixed soil continues to decrease, the cohesion keeps rising, and the strength decreases.

Clay is positively charged, while bacteria are negatively charged. For group NLC0~NLC21 samples treated by MICP, when the clay content is low, the amount of positive charge is small, only a small number of bacteria can be absorbed, and less calcium carbonate can be generated. The contact between the particles is mainly sand particles and sand particles. At this time, the clay fills the pore between the sand particles but fails to fill, so it can be seen that the cohesion and internal friction angle both increase when the clay content is less than 9%. As the clay content gradually increases, the positive charge of the sample increases and the number of bacteria adsorbed increases, which leads to the increase in calcium carbonate production, the pore between the sand particles is filled gradually. The main contact form between the sand particles and calcium carbonate is the contact between the sand particles and the internal friction angle increases accordingly. The clay content continues to increase (over 9%) and when the pores between the sand particles are filled, the sand particles are encapsulated, and the calcium carbonate generated begins to decrease (Figure 12). Because the bacterial solution and nutrient solution will bring excessive clay to the bottom of the sample (inject the liquid from top to bottom), the clay content at the bottom is too high and the pore is reduced, which is not conducive to the survival of bacteria. Moreover, due to the barrier function of clay particles, the sand particles do not contact with calcium carbonate very well. With the increase of clay content, the barrier effect becomes more and more obvious, and the contact between sand particles and calcium carbonate becomes more and more difficult. At this time, the contact form becomes the contact between clay particles and calcium carbonate. Sliding failure of the specimens occurs when subjected to certain external loads (Figure 13), which indicates that the cohesion and internal friction angle gradually decrease with the increase in the clay content.

For further microscopic analysis, the cured columns with different clay contents are magnified 100 times, as shown in Figure 14. The amount of calcium carbonate produced at the different clay contents differs. The amount of calcium carbonate produced by NLC21 groups is small, and the particle image looks smooth (compared with that of the NLC0 group). This result is due to the excessive amount of clay that surrounds the surface of the sand particles, resulting in a smooth SEM image. MICP curing causes the continuous accumulation of calcium carbonate. Initially, the distribution of calcium carbonate in the sand column is disordered. As the reaction progresses, the calcium carbonate accumulates constantly, thus forming particle clusters. For the NLC3 and NLC9 groups in the image, when the clay content is 3% and 9%, the calcium carbonate can accumulate continuously to form particle clusters. When the clay content is 21%, the distribution of the calcium carbonate is disordered because a low clay content can provide an attachment point and survival space for bacteria and can combine with calcium ions continuously to form calcium carbonate, which can generate particle clusters. However, when the clay content is too high, the pores are reduced, the living space of bacteria is destroyed, and calcium carbonate cannot aggregate into clusters, resulting in minimal generation of calcium carbonate, which reduces the shear strength of sand samples. In addition, the sand particles are encapsulated by excessive clay particles, and when the sample is subjected to external loads, the sliding failure of the sand and clay particles also reduces the strength of the sand sample.

## 5. Conclusions

(1)The failure strain and unconfined compressive strength of the MICP-treated sand column increase initially and then decrease with the increase in clay content and reach the maximum when the clay content is 9%. A low clay content is beneficial to the microbial curing effect, the failure strain and unconfined compressive strength of the sample are significantly improved, and the integrity of the sample remains good after damage. A high clay content is not conducive to MICP sand curing, and brittle failure occurs in the sample;(2)With the increase in clay content in the sand–clay mixed soil, the peak strength of the sample under the same confining pressure gradually decreases, the internal friction angle of the mixed soil gradually decreases, and the cohesion gradually increases. After MICP treatment, the internal friction angle and cohesion of sand increase initially and then decrease with the increase in clay content, and the shear strength reaches the maximum when the clay content is 9%. The main reason is that a low clay content can promote the formation of calcium carbonate, enhance the mechanical biting action between particles, and increase the strength of samples;(3)When the clay content exceeds 9%, excessive clay will flow with the bacterial liquid and nutrient liquid, then accumulate at the bottom of the sand sample, which is not conducive to the growth of bacteria and the formation of calcium carbonate. At last, less calcium carbonate will be produced, and the content of available calcium carbonate will be low, finally resulting in low strength;(4)The influence of clay on mixed soil is mainly related to the soil matrix film. When clay and sand particles are mixed, the clay particles wrap around the sand particles to form a soil matrix film with a certain thickness, and this changes the contact form between sand particles, thus affecting the internal friction angle, cohesion, and strength of samples.

## Figures and Tables

**Figure 1 materials-15-03414-f001:**
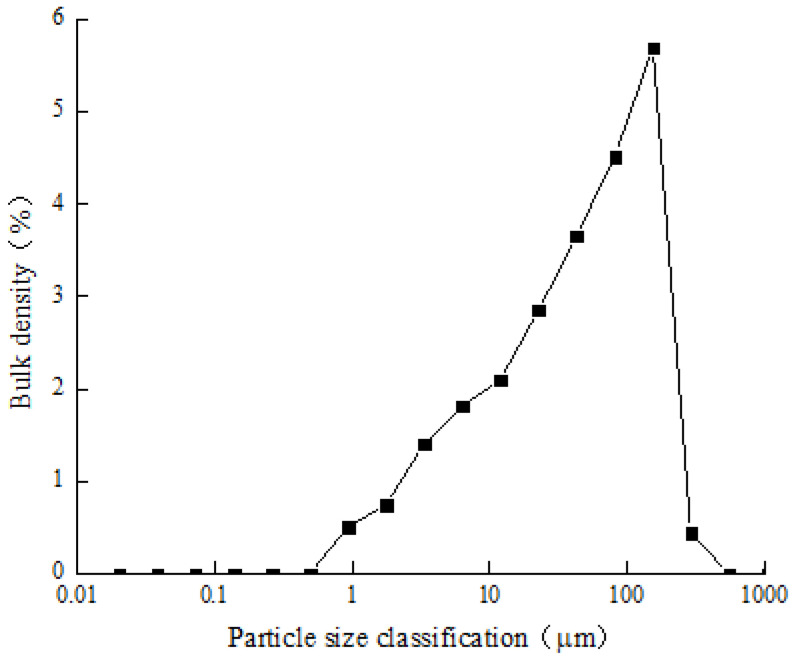
The grain-size distribution of clay.

**Figure 2 materials-15-03414-f002:**
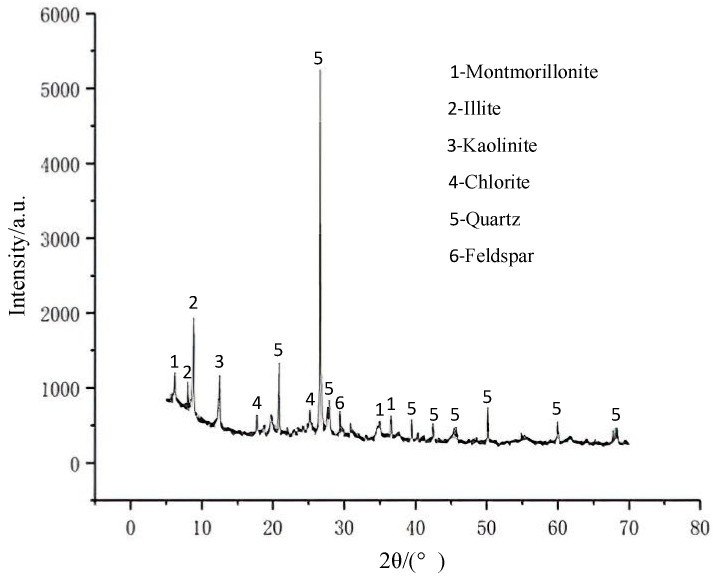
X-ray diffraction pattern of clay.

**Figure 3 materials-15-03414-f003:**
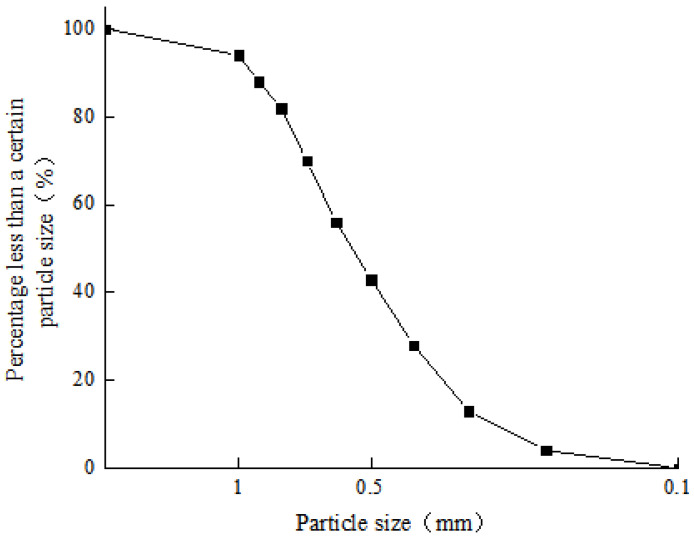
Gradation curve of Fujian standard sand.

**Figure 4 materials-15-03414-f004:**
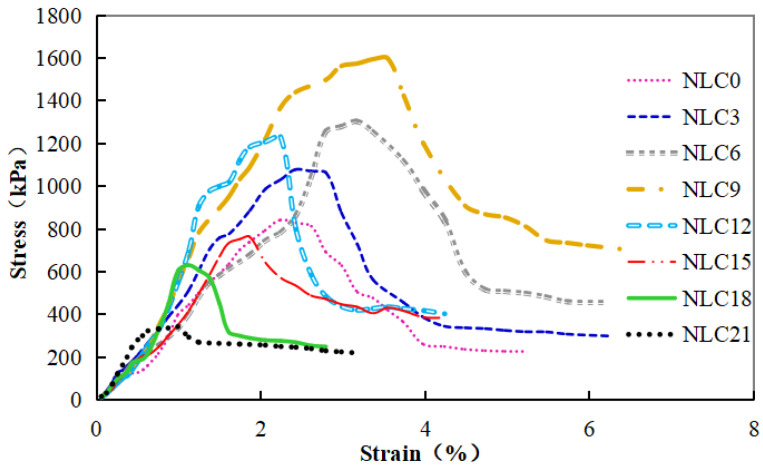
Stress-strain curves of MICP treated sand with different clay content.

**Figure 5 materials-15-03414-f005:**
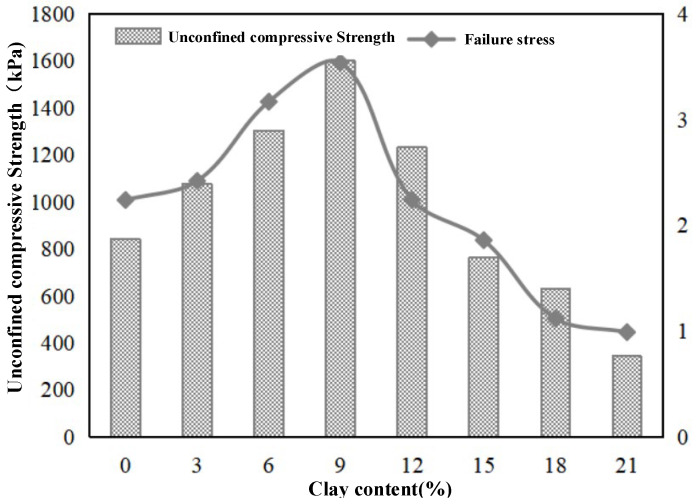
UCS and failure strain with different clay contents (MICP-Treat).

**Figure 6 materials-15-03414-f006:**
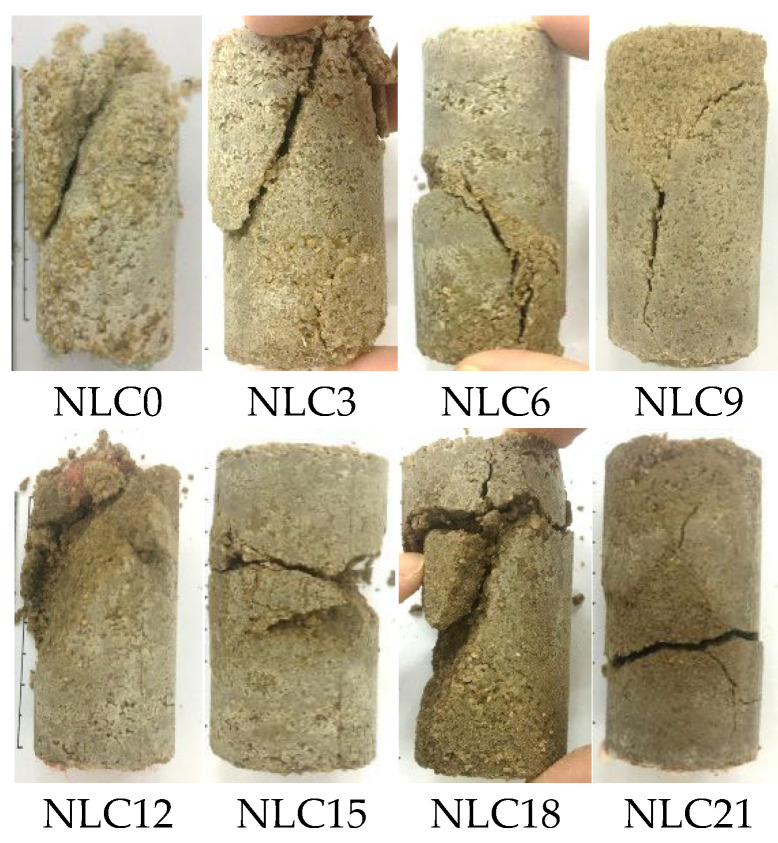
Destruction pattern with different clay contents (MICP-Treat).

**Figure 7 materials-15-03414-f007:**
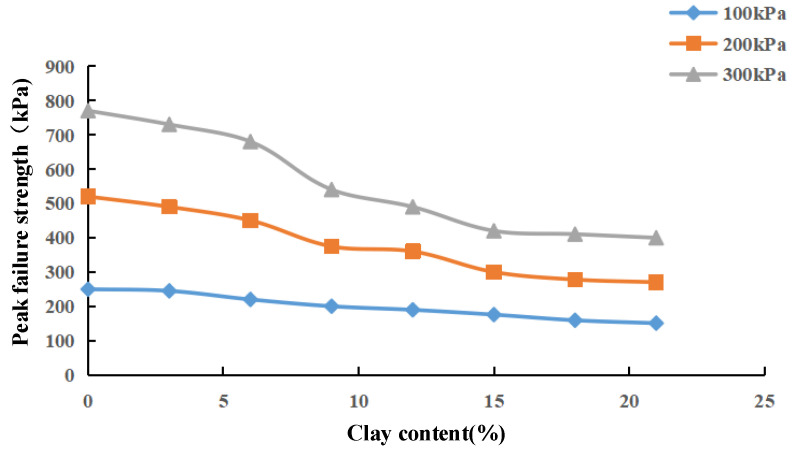
Peak breaking point strength curve with different viscous particle contents.

**Figure 8 materials-15-03414-f008:**
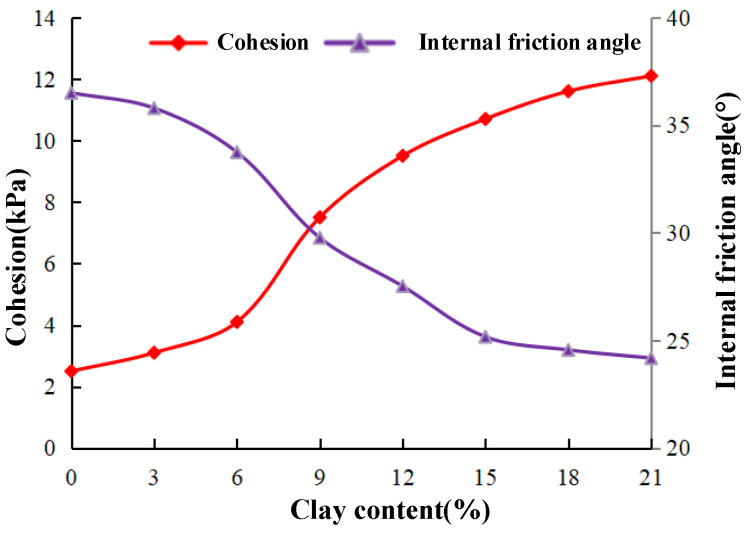
Cohesion and internal friction angle with different clay contents.

**Figure 9 materials-15-03414-f009:**
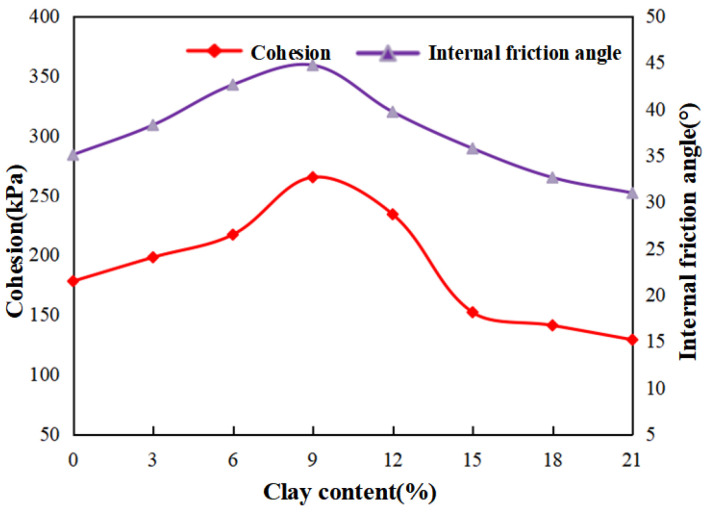
Cohesion and internal friction angle with different clay contents (MICP-Treat).

**Figure 10 materials-15-03414-f010:**
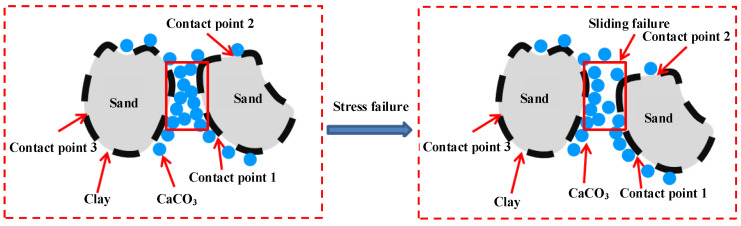
Diagram of sand-clay-CaCO_3_ contact.

**Figure 11 materials-15-03414-f011:**
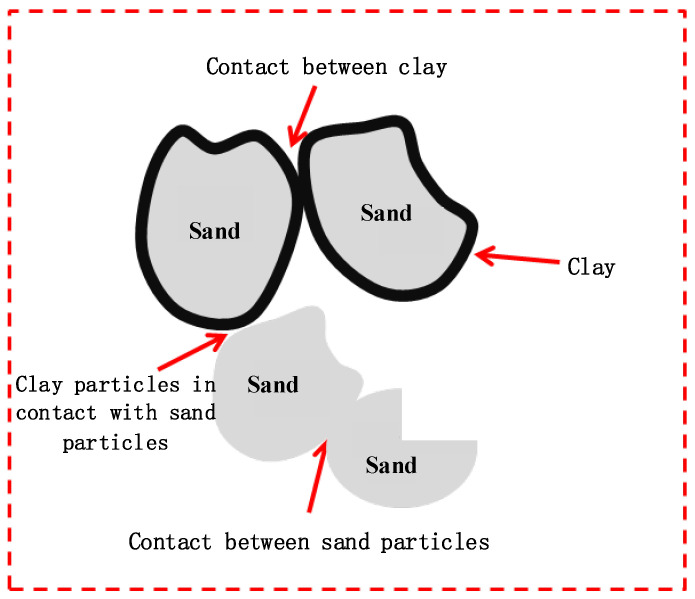
Diagram of sand-clay contact.

**Figure 12 materials-15-03414-f012:**
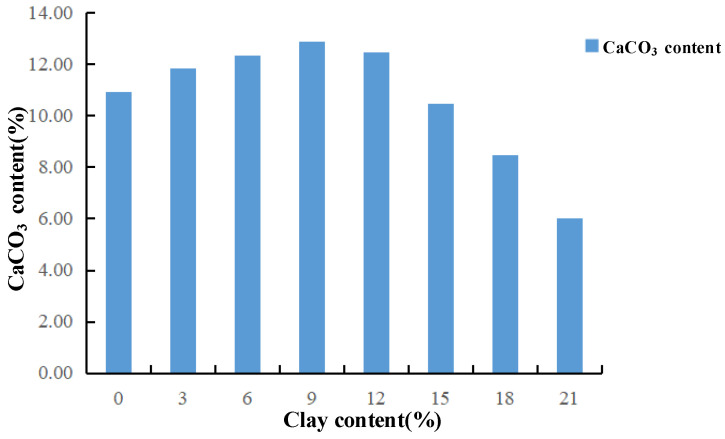
Average calcium carbonate content with different clay contents (MICP-Treat).

**Figure 13 materials-15-03414-f013:**
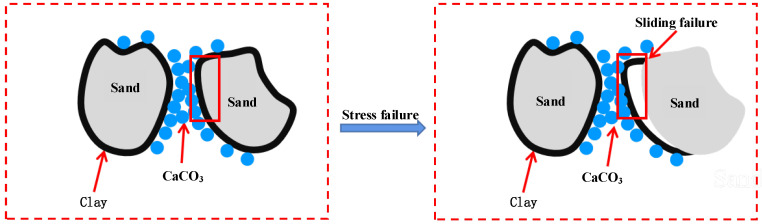
Sand-clay-CaCO_3_ contact diagram.

**Figure 14 materials-15-03414-f014:**
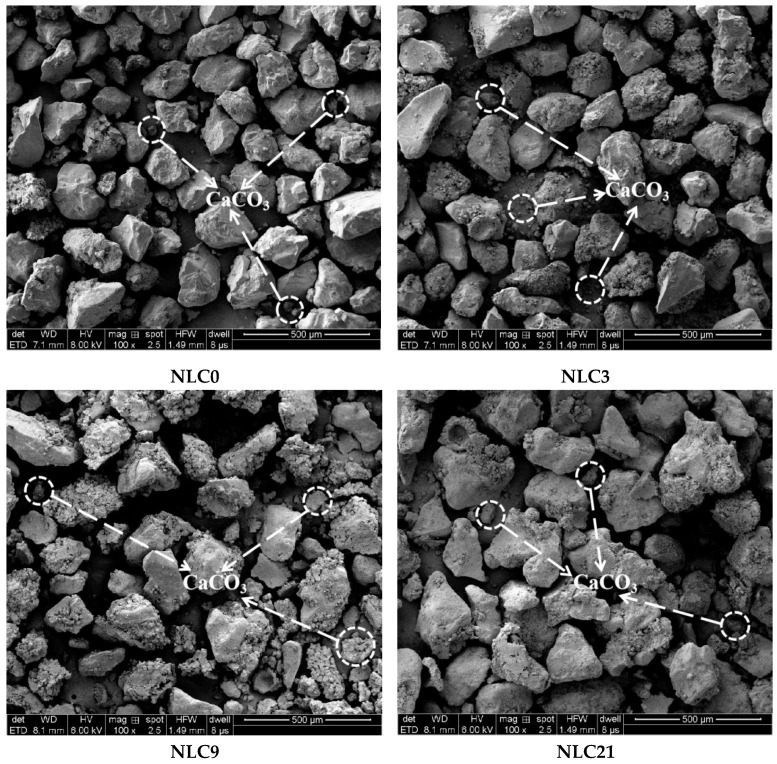
SEM image with different clay contents (100×).

**Table 1 materials-15-03414-t001:** Basic physical indicators of clay.

Moisture Content/%	Density/g·cm^−3^	Void Ratio	Liquid Limit/%	Plastic Limit/%	Liquidity Index	Plasticity Index
69.28	1.67	1.74	52.84	38.14	2.12	14.7

**Table 2 materials-15-03414-t002:** Physical parameters of Fujian standard sand.

Parameter	Density/g·cm^−3^	Specific Gravity	Average Particle Size/(mm)	Nonuniformity Coefficient	Maximum Void Ratio	Minimum Void Ratio
Numerical value	1.92	2.653	0.68	1.56	2.00	0.08

**Table 3 materials-15-03414-t003:** Different clay content test schemes.

Number	Clay Content	Whether Processed by MICP or Not	Consolidated Undrained Test	Unconfined Compressive Strength Test
NL0	0%	No	Confining pressures are 100, 200, and 300 kPa	No
NL3	3%	No	Confining pressures are 100, 200, and 300 kPa	No
NL6	6%	No	Confining pressures are 100, 200, and 300 kPa	No
NL9	9%	No	Confining pressures are 100, 200, and 300 kPa	No
NL12	12%	No	Confining pressures are 100, 200, and 300 kPa	No
NL15	15%	No	Confining pressures are 100, 200, and 300 kPa	No
NL18	18%	No	Confining pressures are 100, 200, and 300 kPa	No
NL21	21%	No	Confining pressures are 100, 200, and 300 kPa	No
NLC0	0%	Yes	Confining pressures are 100, 200, and 400 kPa	Yes
NLC3	3%	Yes	Confining pressures are 100, 200, and 400 kPa	Yes
NLC6	6%	Yes	Confining pressures are 100, 200, and 400 kPa	Yes
NLC9	9%	Yes	Confining pressures are 100, 200, and 400 kPa	Yes
NLC12	12%	Yes	Confining pressures are 100, 200, and 400 kPa	Yes
NLC15	15%	Yes	Confining pressures are 100, 200, and 400 kPa	Yes
NLC18	18%	Yes	Confining pressures are 100, 200, and 400 kPa	Yes
NLC21	21%	Yes	Confining pressures are 100, 200, and 400 kPa	Yes

## Data Availability

The data used to support the findings of this study are included within the article.

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
