# Peer review of "The Effect of Clay on the Shear Strength of Microbially Cured Sand Particles"

_materials, 2022, doi:10.3390/ma15103414_

Round 1
Reviewer 1 Report
The Authors investigated the influence mechanism of different clay contents on the microbial curing effect of sand based on changes in cohesion and friction angle. The laboratory testing was comprehensive. The study is unique and it will benefit the practical engineers. Some English editing are required in the manuscript. Some major revisions are required to improve the quality of the manuscript.
- Change the term "at home and abroad" in line 93 into "in the world"
- Please define the term MICP in line 19 since this is the first time this term introduce.
- Change the term "Scholars" in the manuscript into "Researchers".
- The study include the effect of clay content which also affected the water content of soil. It means there is a change in the unsaturated behavior of the soil. Although the unsaturaed is not part of the study, but the Authors should review the following literatures on the effect of changes in water content due to the unsaturated soil conditions of soil and justify why the unsaturated was not considered in the study.
https://doi.org/10.1016/j.enggeo.2021.106034
https://doi.org/10.1007/s11440-020-01013-8 - What is the meaning of indoor test in line 112?
- Please provide grain-size distribution of clay in the manuscript.
What is the mineral component of clay? IS it expansive clay?
Provide USCS classification of clay and sand - In geotechnical engineering, there is a term plasticity index (PI) which is Liquid limit - plastic limit. What are the definition of liquid limit index and plastic limit index in Table 1?
- Provide density and water content of sand used in the study in Table 2
- Please improve the quality of figure 1
Reviewer 2 Report
Dear authors,
in my opinion, there are some weak points in your paper that should be clarified, improved, and corrected, so that the evaluated work is more understandable and better in perception.
- Title: Is it possible to write it simpler, after all, it is about clay fraction/particles, and their content in clay sands, right?
- The entire paper is saturated with long, convoluted sentences that are difficult to understand. Ex. Abstract - lines 18-21, Introduction - lines 71-73, 87-92, etc. These are just examples. I encourage the authors to work on the text again and improve it.
- Abstract: line 11. It is only in the case of clayey or loamy sands that one can speak of any content of a fine, clay fraction < 0.002 mm. This should be underlined. I encourage the authors to check the granulometric composition of sands.
- Introduction: There are many passages that repeat themselves. e.g. beginning of paragraph 2, beginning of paragraph 5. The sentence: "Scholars at home and abroad have sth..." is repeated too many times.
- Introduction: line 98. Does clay or silt reduce the shear strength? Clay and silt are not the same soil type. I don't understand why the notation "clay (silt)". And further in line 99 "... reduces the shear strength of silt?" Please clarify.
- Introduction: line 100. This should be written in a new paragraph.
- Test materials and schemes. Lines 109-110. Unintelligible sentence. Isn't it about soil taken from the depth ....? Moreover, why clay was taken for testing? If this is about sands. Please clarify. Maybe the authors are talking about some new mixtures? Unfortunately in the article, there was no word about creating mixtures, which should be described in the preparation of the test material.
- Test materials and schemes. Line 110. "The clay particle is fine and has strong viscosity." This is obvious. Unless the authors are referring to clay with admixtures, then it is worth writing the granulometric composition.
- Test materials and schemes. Line 112. "An indoor test". What kind of tests are you referring to? The physical properties of clay listed in Table 1 are investigated by various tests.
- Test materials and schemes. Line 115. " The physical characteristics or the physical parameters". Please decide and change this sentence.
- Test materials and schemes. In fig. 1 is shown the size (or particle) distribution curve.
- Test materials and schemes. Table 2. What is "proportion"? What is this physical parameter?
- Test materials and schemes. Table 3. The caption should be different. this is about sands with varying clay content, right? Moreover, what does CU alone mean here?
- Analysis of test results. Please change "In fig. 2 the stress-strain curves are shown", etc. Please use passive voice, or that the authors presented sth in fig...
- Analysis of test results. In lines 150 and 159, please change the word "manifestations" for a better one. Line 168, what does it means "worsens"? Isn't it about "getting worse"?
- Analysis of test results. Figure 2. When printing in black and white, it is not clear which line is which. The caption should be clearer. After all, this is about sands with varying clay content, right?
- Analysis of test results. Lines 197-198. Please improve this sentence to make it clearer.
- Analysis of test results. Line 226. " The peak failure point strength" is nothing more than the maximum stress deviator at which failure of a specimen occurs, right?
- Analysis of test results. Lines 234 and 235. How does the angle decrease, and the cohesion increase? How much?
- Analysis of test results. The description of subsections 3.4 and 3.5 is the same. CU - consolidated undrained triaxial test. Please change, or combine, subsections 3.4. and 3.5.
- Analysis of test results. Figures 6 and 7 can be put together? The comparison will be more visible. Moreover, the second line should not be an internal friction angle?
- Analysis of test results. Lines 257-259 - strange, incomprehensible sentence.
- Mechanism analysis of the influence of clay content on microbial curing. Lines 280-283 - obvious. Lines 308-318 - are not very innovative.
- Mechanism analysis of the influence of clay content on microbial curing. Figures 8 and 9 - the caption is the same.
- Mechanism analysis of the influence of clay content on microbial curing. Failure to include Figures 10 and 11 in the text of the article, the lack of their analysis. Was this removed by accident or was it forgotten?
- Mechanism analysis of the influence of clay content on microbial curing. Figure 12 - When printing in black and white, nothing is visible here. There is a lack of description, maybe a legend.
- Conclusions. No 4 - too obvious. It should be further considered why a clay fraction content of more than 9% in sand causes a decrease in strength.
Round 2
Reviewer 1 Report
The Authors have revised the manuscript significantly.
The manuscript can be accepted as is now
Reviewer 2 Report
Dear authors. Thank you for answering the questions and making the changes to your manuscript. For the most part, questionable issues have been resolved. Of course, there are still some errors in style, punctuation, etc., which will probably be fixed during proofreading in the journal itself. In my opinion, the article in the presented form, after corrections were done, can be submitted for publication.